analytical chemistry

As(III) detection, paper sensors, smartphone, fluorescence sensors

**Author for correspondence:**
Yong Li
e-mail: liyong07@126.com

# Portable smartphone-integrated paper sensors for fluorescence detection of As(III) in groundwater

Sha Liu, Yong Li, Chao Yang, Liqiang Lu, Yulun Nie and Xike Tian

Faculty of Materials Science and Chemistry, China University of Geosciences, Wuhan 430074, People's Republic of China

(iD) YL, 0000-0003-1295-3856

Arsenic contamination in groundwater is a supreme environmental problem, and levels of this toxic metalloid must be strictly monitored by a portable, sensitive and selective analytical device. Herein, a new system of smartphone-integrated paper sensors with Cu nanoclusters was established for the effective detection of As(III) in groundwater. For the integration system, the fluorescence emissive peak of Cu nanoclusters at 600 nm decreased gradually with increasing As(III) addition. Meanwhile, the fluorescence colour also changed from orange to colourless, and the detection limit was determined as 2.93 nM (0.22 ppb) in a wide detection range. The interfering ions also cannot influence the detection selectivity of As(III). Furthermore, the portable paper sensors based on Cu nanoclusters were fabricated for visual detection of As(III) in groundwater. The quantitative determination of As(III) in natural groundwater has also been accomplished with the aid of a common smartphone. Our work has provided a portable and on-site detection technique toward As(III) in groundwater with high sensitivity and selectivity.

This article has been edited by the Royal Society of Chemistry, including the commissioning, peer review process and editorial aspects up to the point of acceptance.

# 1. Introduction

Arsenic pollutants are mainly introduced into groundwater through sulphide ores mining, industrial operations and agricultural activities [1–4]. Superfluous arsenic in groundwater has induced a serious threat to global public health security. It is reported that arsenic-contaminated groundwater was used as drinking water sources by about 40 million people in the world, which would lead to arsenicosis and liver, bladder and lung carcinomas, etc. [5–7]. Extensive concerns over arsenic exposure caused the World Health Organization (WHO) to lower the

permitted level of arsenic in groundwater to 10 ppb in 2001 [8]. In the contaminated anoxic groundwater, arsenic specifications often occur as arsenite [As(III)] and arsenate [As(V)]. Of the two, As(III) is more stable and toxic, and predominated. The frequently used As(III) detection methods include the traditional Gutzeit method, chromatography, electrochemical, mass spectrometry and biosensing techniques. However, they generally need trained technicians to operate the highly toxic chemical agents or complicated laboratory equipment, which increase the analysis costs and time [9–13]. It would be possible to address this issue by means of an easy-to-use detection method.

From the practical perspective, to develop portable analytical devices for on-site detection of As(III) is highly desired. Paper, a kind of pervasive material in our daily life, has extensively been regarded as promising support owing to its unique portability, fast response, lightweight, flexibility and cost-effective features [14–17]. Flexible sensors with paper as support have been widely investigated, but the issue of detection quantification still needs to be resolved. In recent years, increasing attention has been paid to developing the smartphone-based, low-cost and portable analytical device as a sensing platform. As known, smartphones are extensively employed as a communication tool. But, they can be also equipped with high-speed computing capability, high-performance camera, USB ports and ambient light sensors. With the aid of a colour detection app and the high-performance camera of a smartphone, the intensities of optical signals, such as red (R), green (G) and blue (B), of the paper sensors can be obtained immediately [18–20]. Then, the paper sensors assisted by smartphones have potentials as on-site optical analysis terminals for qualitative and quantitative detection of trace analytes. Zhang *et al.* prepared the mesoporous carbon dispersed Pd nanoparticles-based paper sensors for colorimetric detection of $H_2O_2$ [21]. Zhao *et al.* also developed the ambient light-based smartphone platform for the simultaneous detection and absorption of organophosphorus pesticides [22]. Comparing with the common ultraviolet optical signal, fluorescence may be a more suitable candidate due to its higher sensitivity, selectivity, shorter analysis time and clearer visual ability [23–25]. Over the past decades, extensive studies have been performed for fluorescence detection of micropollutants. The non-noble metal Cu nanoclusters can avoid the shortcoming of quantum dots and gold, silver nanoclusters, then the study of Cu nanoclusters has attracted extensive attention due to their earth-abundance, lower cost, outstanding fluorescence property, easy-to-use decoration, etc. [26–28]. Polyethyleneimine-templated Cu nanoclusters from ascorbic acid reduction were applied for $Fe^{3+}$ detection [29]. One-step synthesis of L-cysteine stabilized Cu nanoclusters has been used as fluorescent probes for $Hg^{2+}$ and $S^{2-}$ detection [30]. Although Cu nanoclusters are widely investigated for micropollutants detection, studies on As(III) detection have never been reported. In view of this, the fluorescence paper sensors based on Cu nanoclusters assisted by a smartphone would be an excellent portable analytical device for field detection of As(III) in groundwater.

Herein, the water-soluble Cu nanoclusters were prepared via a facile reduction method. The Cu nanoclusters can emit intense orange fluorescence with a quantum yield of 14.6%. And also, they can detect As(III) with the detection limit of 2.93 nM in a wide linear concentration range. As(III) detection by Cu nanoclusters is based on the strong interaction between As(III) and glutathione onto the surface of nanoclusters to induce the intense fluorescence quenching. The common ions in water have no interferences on the sensitive detection of As(III), indicating their high and efficient detection selectivity. The portable paper sensors based on Cu nanoclusters have been fabricated by an easy-to-use method for portable detection of As(III). The fluorescence colours of paper sensors upon spiked As(III) in groundwater can be recorded and analysed by a colour detection app and a smartphone. With this regard, a portable integration system has been founded for field detection of As(III) in groundwater with high sensitivity and selectivity (scheme 1).

# 2. Experimental

## 2.1. Synthesis of water-soluble Cu nanoclusters

First, 12.50 mg of copper nitrate in 5 ml of water and 0.25 g of glutathione were dissolved and mixed in 10 ml of water with stirring for 10 min. Then, 10 µl of 0.01 mol l$^{-1}$ sodium hydroxide was added dropwise. The solution colour changed from turbid to yellow. After stirring vigorously for 1 h at 65°C, the prepared Cu nanoclusters were filtrated by a 0.22 µm membrane three times and purified by transferring into a dialysis bag (MW = 500). Finally, Cu nanoclusters were freeze-dried and dispersed in BR buffer solution for further use.

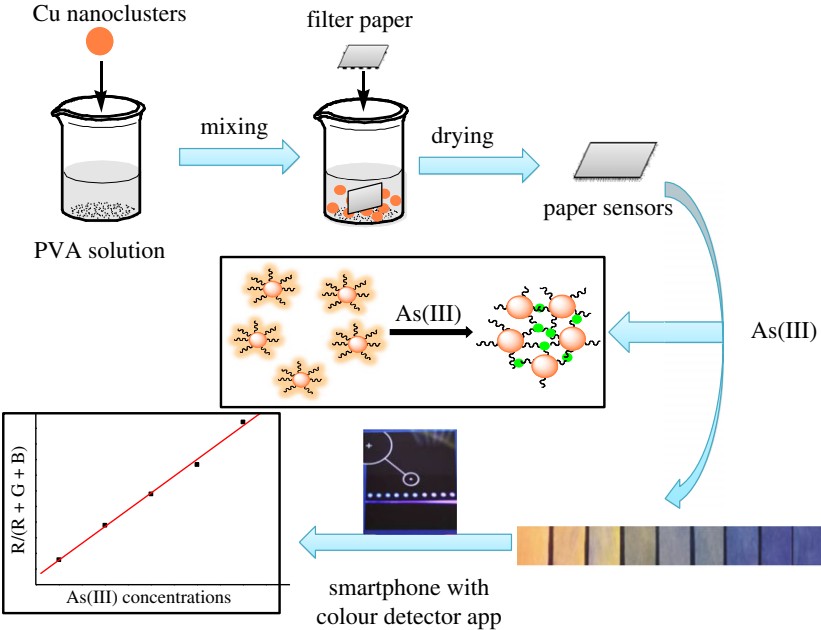

**Scheme 1.** Illustration for the smartphone-assisted paper sensors with Cu nanoclusters for As(III) detection.

## 2.2. Sensitive and selective detection of As(III) by Cu nanoclusters

Sensitivity is an important criterion to assess the applicability of a sensor. Generally, 2 ml of Cu nanoclusters BR buffer dispersion was added into a quartz cuvette. After that, 20 µl of $1 \times 10^{-5}$ mol l$^{-1}$ As(III) stock solution was added successively. Their emission spectra were recorded from 470 to 720 nm with an excitation of 380 nm with a spectrophotometer. The natural groundwater usually contains many interfering anions and cations; hence, the selective detection property of Cu nanoclusters on As(III) should be studied. The fluorescent titration of Cu nanoclusters with interfering anions, such as $NO_3^-$, $CO_3^{2-}$, $HCO_3^-$, $SO_4^{2-}$, $C_2O_4^{2-}$, $PO_4^{3-}$, $BrO_3^-$, $ClO_4^-$, $F^-$, $Cl^-$, $Br^-$ and $I^-$ and interfering cations, such as $Na^+$, $K^+$, $Al^{3+}$, $Fe^{3+}$, $Fe^{2+}$, $Cu^{2+}$, $Zn^{2+}$, $Cd^{2+}$, $Hg^{2+}$, $Ca^{2+}$, $Mg^{2+}$ and $Pb^{2+}$ was conducted by the same procedure used for As(III) detection. The fluorescence competitive selective experiments were also conducted. An aliquot of 10 µM interfering anions or cations was mixed with Cu nanoclusters in 2.0 ml of BR buffer solution, respectively, and then 0.1 µM of As(III) was added into the spectrophotometer quartz cuvette. The fluorescence spectra were subsequently recorded. All measurements were performed at least in triplicate at ambient conditions.

The reliability and practicality of Cu nanoclusters on As(III) detection were investigated in natural water samples spiked with different concentrations of As(III). The recovery test was performed in real water samples, such as ultrapure water, tap water and groundwater from the Jianghan Plain. All water samples were used after simple pretreatment (high-speed centrifugation at 10 000 r.p.m. for 5 min to remove a small number of sediments, 0.45 µm filter filtration and pH adjustment), and then the real water samples were spiked with different concentrations of As(III) (50, 100 and 150 nM). Each concentration of As(III) was tested in triplicate, and the average value was presented with standard deviation.

## 2.3. Paper sensors fabrication and their application on detection of As(III) in groundwater

Paper sensors based on Cu nanoclusters were fabricated as below. First, 1 g of poly (vinyl alcohol) (PVA-1788) was dissolved in 20 ml of water. Then, 1 ml of Cu nanoclusters solution (5 µg ml$^{-1}$) was added and stirred vigorously for 15 min. The common filter paper was dropped into the mixture solution for 30 s and then taken out. The paper sensors were then dried at room temperature for 30 min and cut into strips ($1 \times 5$ cm) for use. Smartphone-integrated paper sensors for As(III) detection was conducted. Twenty microlitres of As(III) stock solutions with concentrations of 0, 0.15, 0.30, 0.45, 0.60, 0.75, 0.90, 1.05 and 1.20 µM were added into 1.98 ml of BR buffer solution. Then, the fabricated paper sensors were immersed into the above mixture solutions. After standing for 20 s, the paper sensors were taken out and placed under a UV lamp to shoot their fluorescence colours. Thus, the loading amount of Cu nanoclusters on the paper can be regulated effectively by controlling the nanocluster concentrations,

immersing time etc.; then, fluorescence colour from the paper sensors can be strictly controlled. The fluorescence colour of paper sensors under a UV lamp was analysed by a smartphone equipped with a colour detector app. The smartphone was placed in a holder at a fixed distance (20 cm) for colour scanning, and the colour parameters (R, G and B) can be recorded by the app.

In order to evaluate the practical As(III) detection ability of smartphone-integrated paper sensors, As(III) concentrations in the natural groundwater from some typical high-arsenic groundwater areas in China, such as Datong Basin, Xiantao Jianghan Plain, Foshan Pearl River Delta and Kuitun Xinjiang, have been determined. Firstly, all the natural groundwater samples were used after simple pretreatment (high-speed centrifugation at 10 000 r.p.m. for 5 min to remove a small amount of sediments, 0.45 µm filter filtration and pH adjustment). Then, As(III) concentrations in these natural groundwater samples were determined by HPLC-ICP-MS, and our developed smartphone-integrated paper sensors in triplicate and the average values were presented with standard deviation.

# 3. Results and discussion

## 3.1. Preparation and characterization of Cu nanoclusters

The water-soluble Cu nanoclusters were prepared via a simple and facile reduction route. In the synthesis, glutathione acted a crucial role. The thiol group in glutathione is beneficial to form the high-affinity metal-ligand clusters, and their carboxyl and amino groups are necessary to enhance the stability of Cu nanoclusters. In addition, the amino groups in glutathione are also reacted as reductive agents to synthesize Cu nanoclusters. Therefore, glutathione is not only a reductive agent but also a protective agent [31–33], which is greatly helpful for the preparation of Cu nanoclusters with high stability and fluorescence quantum yields. The selected reaction parameters, such as reaction time, temperature, pH, glutathione concentration and copper precursors, pose great influences on Cu nanoclusters preparation. To figure out the optimal synthetic conditions, a series of control experiments were performed. The fluorescence intensity of Cu nanoclusters increased gradually with reaction time prolonging till 60 min. If the reaction time continues to increase, the fluorescence intensity would decrease. The optimal reaction time for Cu nanoclusters is chosen as 60 min (electronic supplementary material, figure S1). The reaction temperature is another important parameter to influence the fluorescence intensity. When the reaction temperature is 65°C, the fluorescence ability is strongest (electronic supplementary material, figure S2). Furthermore, glutathione concentrations and pH values of the reaction solution and copper precursors would also greatly influence the fluorescence intensities of Cu nanoclusters (electronic supplementary material, figure S3–S5). Thus, the reaction time of 60 min, reaction temperature of 65°C, pH of 4.0 and copper nitrate and glutathione concentration of 50 mg ml$^{-1}$ served as the optimal synthetic conditions for Cu nanoclusters.

The prepared Cu nanoclusters have an enhanced fluorescence quantum yield of 14.6% with quinoline sulphate as the reference [34]. Also, their optical property, structure and morphology were systematically characterized by FTIR, TEM, XPS, etc. Cu nanoclusters can emit intense orange fluorescence with the emission of 600 nm under a single excitation of 380 nm (electronic supplementary material, figure S6). Under sunlight, the solid powder of Cu nanoclusters is colourless, and its water solution is transparent. Under UV light, the solid powder of Cu nanoclusters exhibited orange fluorescence, and its water solution also can emit intense orange fluorescence. While no fluorescence can be observed for glutathione and $Cu(NO_3)_2$, indicating the fluorescence of the nanoclusters is originated from Cu (electronic supplementary material, figure S7). As shown in figure 1a, the peak at 2525 cm$^{-1}$ in FTIR spectra of glutathione can be assigned to S-H vibration. This peak disappeared for Cu nanoclusters, indicating that glutathione was successfully decorated onto Cu nanoclusters [35]. TEM image was obtained to study the morphology and particle size distributions of Cu nanoclusters. As shown in figure 1b, the prepared Cu nanoparticles are spherical with about 3.0 ± 0.2 nm and finely disseminated with no clear aggregations. The data of size distribution analysis of Cu nanoclusters from Malvern potential instrument are consistent with the TEM analysis (electronic supplementary material, figure S8). XPS spectra were performed to gain insights into the components of Cu nanoclusters. The six main components of Cu nanoclusters are C, N, O, S, Cu and Na. The intensity of the Cu 2p peak is weak due to the low concentrations of Cu nanoclusters in water (figure 1c). As shown in figure 1d, two intense peaks can be assigned to the amplified peak of Cu $2p_{3/2}$ and Cu $2p_{1/2}$, which are at 932.05 and 952.05 eV, respectively. In addition, the satellite peak at 942 eV which is the feature peak of $Cu^{2+}$ ions cannot be found, indicating Cu nanoclusters are mainly composed of $Cu^0$ and $Cu^+$ [36].

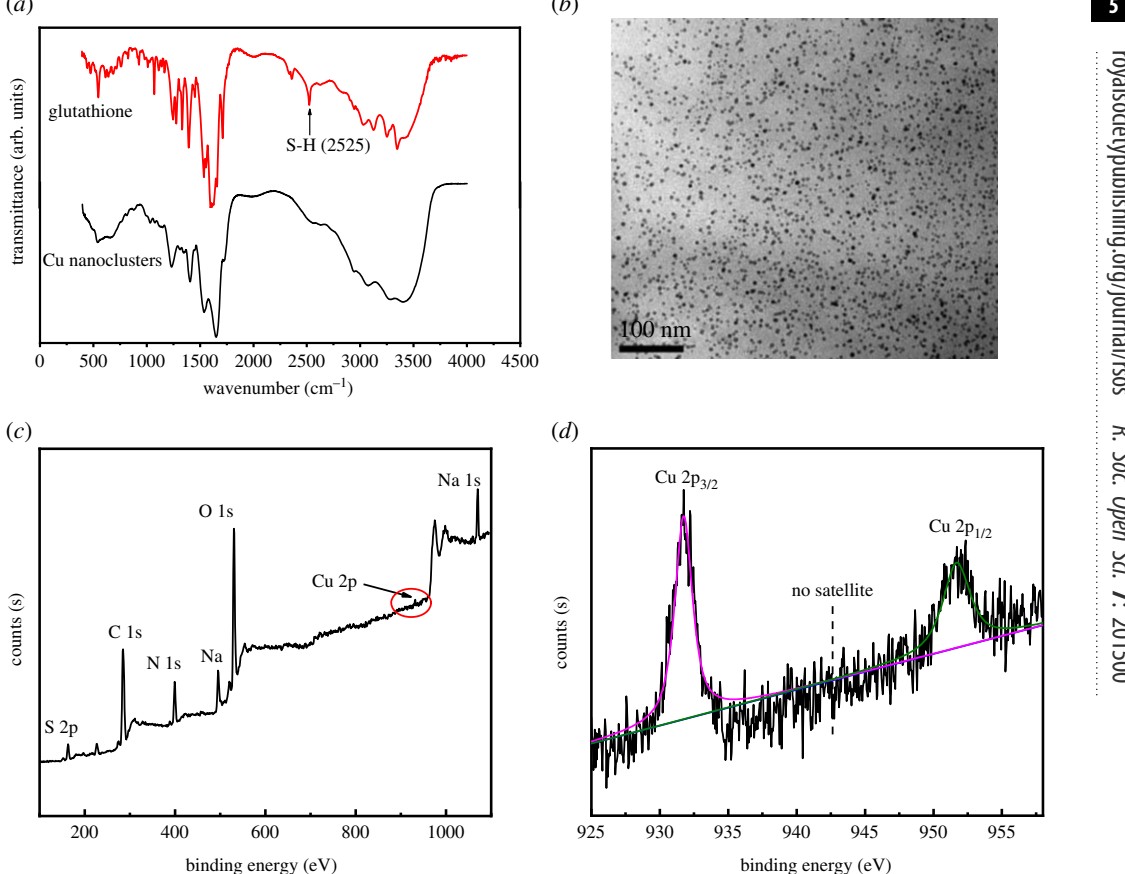

**Figure 1.** (*a*) FTIR spectra of Cu nanoclusters and glutathione, (*b*) TEM image of Cu nanoclusters, (*c*) XPS survey spectra of Cu nanoclusters and (*d*) amplified XPS spectra of Cu2p in Cu nanoclusters; no satellite peaks indicated no existence of $Cu^{2+}$.

## 3.2. Sensitive and selective detection of As(III) by Cu nanoclusters

The fluorescence intensity of Cu nanoclusters at 600 nm dramatically decreased upon the introduction of As(III) in water (electronic supplementary material figure S9). Meanwhile, their fluorescence colour changed to colourless from orange, indicating that the prepared Cu nanoclusters are a potential sensor for As(III) detection. To explore the possible detection mechanism, size distribution and zeta potential analysis were performed. The Cu nanoclusters are negatively surface charged with the zeta potential of −32.3 mV. With the addition of 1.0 and 2.5 μM As(III), the zeta potential values have changed to −42.3 and −51.8 mV. In addition, we noticed that the size of Cu nanoclusters also increased, which indicated that the introduction of As(III) into Cu nanoclusters solution would lead to the aggregations of Cu nanoclusters. In short, As(III) can bind with glutathione via the stable As-S linkage due to the strong interaction between them, and the introduction of As(III) into glutathione-decorated Cu nanoclusters would lead to the instability and aggregation of Cu nanoclusters and finally induced the fluorescence quenching of Cu nanoclusters [37–39].

The As(III) detection ability of Cu nanoclusters is highly dependent on the pH values of the solution. When the pH of BR buffer is 6.0, the fluorescence of Cu nanoclusters quenched most greatly (electronic supplementary material, figure S10). Thus, a pH of 6.0 was selected to study the detection ability of Cu nanoclusters. Under the optimal detection conditions, the fluorescence titrations of Cu nanoclusters toward As(III) of various concentrations (0, 0.1, 0.2, 0.3, 0.4, 0.5, 0.6, 0.7, 0.8, 0.9, 1.0, 1.1, 1.2, 1.3, 1.4, 1.5, 1.6, 1.7, 1.8, 1.9 and 2.0 μM) were performed (figure 2*a*). As shown, Cu nanoclusters also showed an intense fluorescence peak at 600 nm with excitation of 380 nm. Upon the gradual addition of As(III), the fluorescence intensity of Cu nanoclusters decreased gradually, which is based on the strong interaction between As(III) and glutathione onto the surface of nanoclusters. The fluorescence dose-response of Cu nanoclusters with As(III) was examined (figure 2*b*). In figure 2*b*, the *X*-axis represents the concentrations of As(III), which are equal to the number on the *X*-axis multiplied by

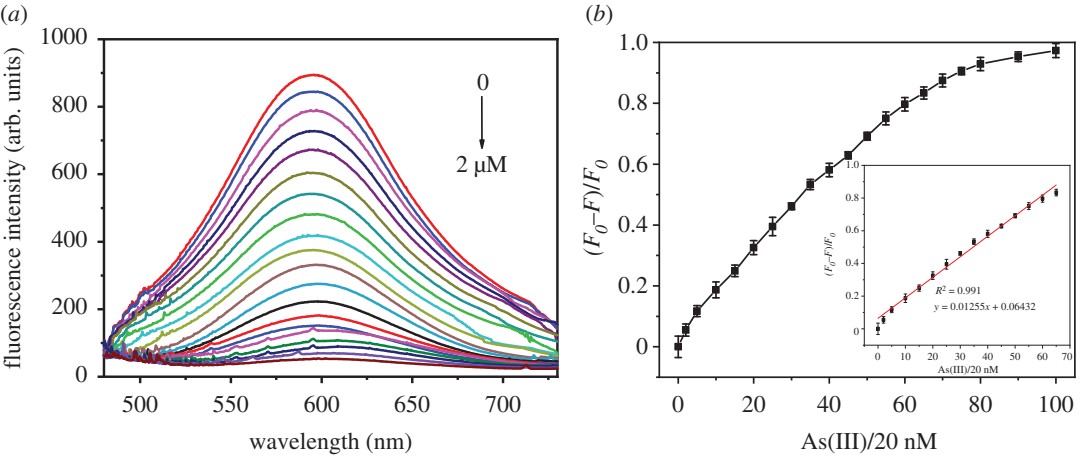

**Figure 2.** (a) Fluorescence spectra of Cu nanoclusters with increasing addition of As(III) (0 to 2.0 μM) under the excitation of 380 nm, all the fluorescence measurements were conducted in BR buffer solution, (b) plot of the fluorescence change $(F_0–F)/F_0$ versus As(III) concentrations (0 to 2 μM), and the inset is the linear plot of fluorescence changes versus As(III) concentrations (0 to 1.3 μM).

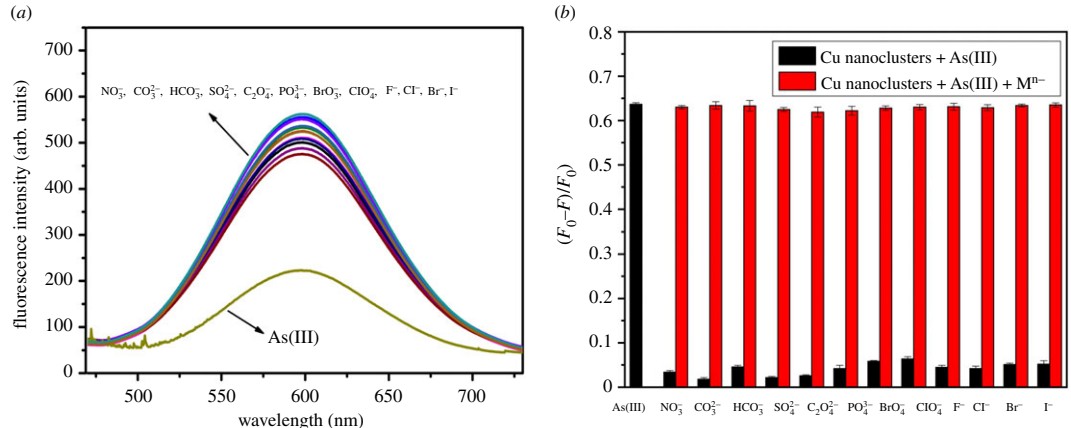

**Figure 3.** (a) Fluorescence spectra of Cu nanoclusters toward different interfering anions (0.5 μM for As(III), 10 μM for the others anions); (b) The selectivity of Cu nanoclusters toward As(III) over other anions. The black bars represent the fluorescence response of Cu nanoclusters toward As(III) (0.5 μM) and other interfering anions (As(III) (10 μM). The red bars represent the fluorescence response of Cu nanoclusters toward interfering anions (10 μM) in the presence of 0.5 μM As(III). All experiments were carried out in BR buffer solution.

20 nM. We then can find that the relative fluorescence intensities $(F_0–F)/F_0$ increase dramatically with increasing As(III) concentrations initially, and it tends to be saturated at a higher concentration. The relative fluorescence intensity has good linearity with As(III) concentrations ranging from 0 to 1.3 μM (inset of figure 2b). The correlation coefficient was as high as 0.991. The detection limit is equal to $3 \times SD/\rho$. In the formula, SD is the standard deviation of blank measurement, and $\rho$ is the slope between the fluorescence intensity ratio versus As(III) concentration. Then, the detection limit of Cu nanoclusters toward As(III) is calculated to be 2.93 nM (0.22 ppb). Comparing with some reported fluorescence sensors and As(III) permitted level in drinking water, the detection limit was much lower (electronic supplementary material, table S1) [40–47]. In summary, As(III) assay based on Cu nanoclusters has displayed favourable linear range and low detection limit.

To investigate the detection selectivity of Cu nanoclusters toward As(III), the fluorescence spectra of Cu nanoclusters in the presence of the interfering anions were studied under the same experimental conditions as As(III). As shown in figure 3a, the fluorescence intensity of Cu nanoclusters in the presence of As(III) at 600 nm decreased greatly. And the fluorescence colour also changed from orange to colourless. When the interfering anions were added into Cu nanoclusters, the fluorescence spectra still kept stable accompanying no fluorescence colour changing, indicating the prepared Cu nanoclusters possess a strong selectivity toward As(III) detection. Furthermore, the competitive selective experiments were also carried out by

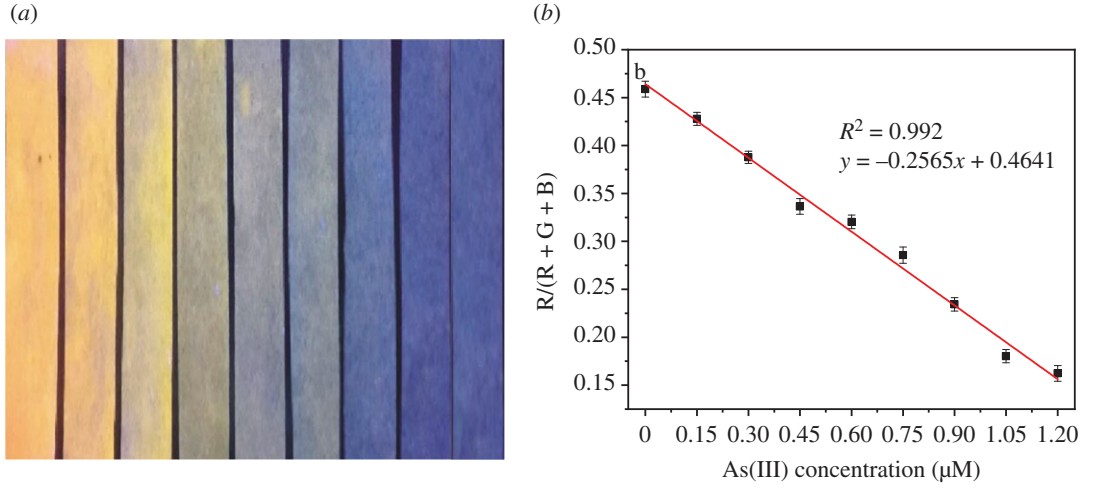

**Figure 4.** (*a*) Fluorescence images of paper sensors upon the increasing addition of As(III) (from left to right: 0, 0.15, 0.30, 0.45, 0.60, 0.75, 0.90, 1.05 and 1.20 µM). All the images were taken under a 365 nm UV lamp; (*b*) The linear relationship between the colour parameters of paper sensors and As(III) concentrations.

adding As(III) into a mixture solution of Cu nanoclusters and other interfering anions. As shown in figure 3*b*, no obvious interferences can be observed, even when the concentrations of the coexisting anions were even 20 times higher than that of As(III) under the optimal detection conditions, indicating that As(III) detection by Cu nanoclusters is hardly affected by these common coexisting anions and the as-prepared Cu nanoclusters have a good selectivity toward As(III) detection over other competitive anions. In addition, we have also evaluated the detection selectivity of Cu nanoclusters over the common cations (electronic supplementary material, figure S11), showing that the common cations also have no great interferences on As(III) detection.

The reliability of As(III) detection by Cu nanoclusters was also evaluated. The real water samples, such as ultrapure water, tap water and groundwater, were first obtained. And, these water samples were pretreated through centrifugation, filtration, etc. to remove all the insoluble impurities. Then, different concentrations of As(III) (50, 100 and 150 nM) were spiked. Each As(III) concentration was determined in triplicate, and the average values were presented with standard deviation (electronic supplementary material, table S2). Experimental results indicated the recovery of As(III) in the three different real water samples fluctuated from 98.2% to 104.2%. Furthermore, the calculated As(III) concentration from Cu nanoclusters was highly consistent with the spiked amount. The reliability test exhibited that Cu nanoclusters can serve as a credible and efficient fluorescence sensor for As(III) screening in real water samples.

## 3.3. As(III) detection by the portable smartphone-assisted paper sensors

As mentioned earlier, portable and on-site visual detection of trace As(III) in groundwater is important for the actual applications. For this purpose, Cu nanoclusters decorated on paper sensors have been developed and used for the detection of As(III) in natural groundwater. The paper sensors were fabricated based on the cellulose paper, which was quite stable in the dark, and no fluorescence degradations can be observed after a long-term storage. As shown in figure 4*a*, the prepared paper sensors based on Cu nanoclusters have shown the bright orange fluorescence in the absence of As(III) under a 365 nm UV light. With increasing As(III) concentrations from 0 to 1.2 µM, the fluorescence colour of paper sensors became darker and darker simultaneously, which is due to the fluorescence quenching induced by the addition of As(III). These results indicated that the paper sensors based on Cu nanoclusters are highly desirable for the rapid and visual detection of As(III) in groundwater. But, the specific concentration of As(III) in groundwater cannot be obtained just from fluorescence colour variations of the paper sensors.

Considering that, a smartphone-integrated system has been established for quantitative As(III) detection. The paper sensors upon different As(III) concentrations can exhibit different fluorescence colour. This colour was then scanned and analysed by colour detector app equipped on a common smartphone. The parameters of a different colour, such as red (R), green (G) and blue (B) can be present [48,49]. The paper sensors we fabricated alone can exhibit orange fluorescence, and then

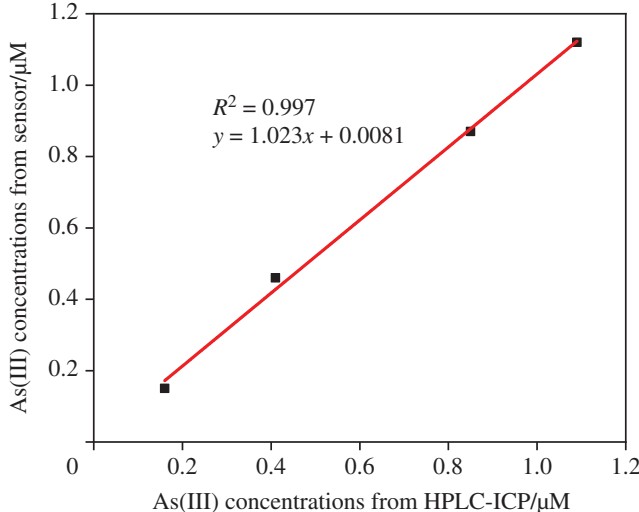

**Figure 5.** Comparisons of HPLC-ICP-MS and smartphone-integrated paper sensors with As(III) concentrations from Datong Basin, Xiantao, Foshan and Kuitun.

gradually changed to colourless with a gradual increase of As(III) concentrations. So, $R/(R + G + B)$ ratio of different colour versus As(III) concentrations was chosen to evaluate fluorescence response. A calibration curve with the coefficient of 0.992 between the fluorescence colour changes and As(III) concentrations ranging from 0 to 1.2 μM was constructed (figure 4b). The lowest detectable concentration of the paper sensor is approximately 92.96 nM (6.98 ppb) based on the detection limit calculation equation. Obviously, the detection limit from the paper sensors is higher than that from the fluorescence detection method. Different factors such as the tight loading of Cu nanoclusters on the paper, colour parameter collection sensitivity, UV lamp power, exposure time, etc. would decrease the detection sensitivity of paper sensors toward As(III) [50,51]. Furthermore, the calculated As(III) concentrations are consistent with the spiked values, indicating the efficient practicality of this detection method. Nevertheless, the method of integrating smartphone and colour detector app with fluorescence paper sensors can complete the visual, on-site and quantitative detection of As(III) in the real groundwater samples.

To evaluate the practical As(III) detection ability of smartphone-integrated paper sensors, the natural groundwater samples from some typical high-arsenic groundwater areas in China, such as Datong basin, Xiantao Jianghan Plain, Foshan Pearl River Delta and Kuitun Xinjiang were collected. First, all the natural groundwater samples were used after centrifugation, filtration and pH adjustment. HPLC-ICP-MS has been extensively used for arsenic speciation analysis [52–54]. Then, As(III) concentrations in these natural groundwater samples were determined by HPLC-ICP-MS and our developed smartphone-integrated paper sensors in triplicate, and the average values were presented with standard deviation. The determined As(III) concentrations by these two different detection methods were listed in electronic supplementary material, table S3. The Pearson correlation coefficient of 0.998 showed a strong correlation. And also, the correlation is significant due to their $p$-value of 0.001. The higher Pearson correlation coefficient and lower $p$-value indicated that the smartphone-integrated paper sensors possessed a similar performance to HPLC-ICP-MS on arsenic speciation determination. Furthermore, As(III) concentrations from HPLC-ICP-MS and smartphone-integrated paper sensors are compared in figure 5. As shown, a good linear curve with a coefficient of 0.997 can be obtained, indicating that As(III) concentrations from smartphone-integrated paper sensors are reliable and can be applied for As(III) detection in the real groundwater samples.

## 4. Conclusion

In summary, a new system of smartphone-integrated paper sensors with Cu nanoclusters was established for the portable and on-site detection of As(III) in groundwater. In the integration system, Cu nanoclusters can emit intense orange fluorescence with a fluorescence quantum yield of 14.6%. Fluorescence titration experiments indicated fluorescence intensity at 600 nm of Cu nanoclusters gradually decreased with increasing addition of As(III) accompanying with fluorescence colour

changing from orange to colourless, which is due to the strong interaction between glutathione and As(III). The detection limit of Cu nanoclusters toward As(III) was determined as 2.93 nM (0.22 ppb) from 0 to 1.3 µM. And the common interfering ions in water have no obvious influences on the selective detection of As(III). The spiked and recovery tests indicated the reliability of Cu nanoclusters for As(III) detection in real water samples. Furthermore, the portable paper sensor based on Cu nanoclusters was developed for the on-site and visual detection of As(III) in groundwater. To determine As(III) concentrations in groundwater accurately, a new system of paper sensors assisted by smartphone has been established. The integration system can be used for the detection of As(III) in groundwater with a low detection limit of 92.96 nM (6.98 ppb), and their practical detection ability in natural groundwater samples was evaluated. In summary, a quantitative method based on Cu nanoclusters-decorated paper sensors assisted by a smartphone has been constructed for the portable, sensitive, selective and on-site analysis of As(III) in groundwater.

Data accessibility. All raw and processed data, analysis scripts for this study are accessible on the Dryad Digital Repository (https://dx.doi.org/10.5061/dryad.m905qftzr [55]).

Competing interest. There are no conflicts to declare.

Funding. This work was supported by the National Natural Science Foundation of China (grant nos. 41807200, 41773126, 51878633) and the Foundation for Innovative Research Groups of the National Natural Science Foundation of China (grant no. 41521001) and the 'Fundamental Research Funds for the Central Universities'.

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
