## [Reviewer comments · Royal Society Open Science]

Review History

RSOS-201500.R0 (Original submission)

Review form: Reviewer 1

Is the manuscript scientifically sound in its present form?

Yes

Are the interpretations and conclusions justified by the results?

No

Is the language acceptable?

No

Do you have any ethical concerns with this paper?

No

Have you any concerns about statistical analyses in this paper?

No

Recommendation?

Major revision is needed (please make suggestions in comments)

Comments to the Author(s)

Manuscript Title: Portable smartphone integrated paper sensors for fluorescence detection of As(III) in groundwater

Manuscript ID: RSOS-201500

The manuscript reports fabrication of Cu nanoclusters and their applicability for As(III) analysis by utilising integrative paper-based fluorescence detection. The authors did a lot of experiments, but there are some points should be clarified.

- 1- The authors mentioned that, "glutathione is not only a reductive agent but also a protective agent,.....", suitable references should be added.
- 2- How the authors control the Cu nanoclusters loaded on the paper? The repeatability of this sensor should be discussed.
- 3- Figure 2b should be corresponding to the variation of As(III) concentrations in Fig.2a, The X axis should be from 0.0 – 2 μ M.
- 4- The concentrations of As(III) should be legend in Fig2a.
- 5- It was found that, the detection limit using fluorescence was 2.93 nM, while increased to 92.96 nM using smartphone APP. The authors should compare the sensitivity in Figs. 2b and Fig. 4b and give explanation.
- 6- It is surprising the proposed method have a good accuracy compared to HPLC-ICP-MS as shown in Fig. 5.
- 7- The manuscript need English revision.

Review form: Reviewer 2

Is the manuscript scientifically sound in its present form?

Yes

Are the interpretations and conclusions justified by the results?

Yes

Is the language acceptable?

Yes

Do you have any ethical concerns with this paper?

No

Have you any concerns about statistical analyses in this paper?

No

Recommendation?

Accept with minor revision (please list in comments)

Comments to the Author(s)

Level of arsenic contamination in groundwater should be strictly monitored due to their close relationship with human health and environmental safety. The manuscript describes a new material for the detection of arsenite with high sensitivity and selectivity. The detection limit of this sensor toward arsenite is quite low with a relatively wide range detection. In addition, the paper sensors were also developed and the optical signal can be quickly read by a common smart phone. Moreover, arsenite concentrations in real water samples can be determined by the as-prepared sensor. In summary, the manuscript can be accepted after minor revisions.

1 What are the advantages of Cu nanoclusters for micropollutants detection, e.g. arsenite detection?

2 In the experimental section, authors should address all of metal salts that they used in this experiment.

3 In the results and discussion, authors reported the size of the prepared Cu nanoparticles are spherical with about 3 nm. Please add standard deviation of the particle size.

4 Please clarify the operation procedure of arsenite determination method by a common smartphone equipped with Color Detector APP.

5 There are some minor errors in grammar and spelling. Authors should check and revise the manuscript more carefully.

Decision letter (RSOS-201500.R0)

Dear Dr Li:

Title: Portable smartphone integrated paper sensors for fluorescence detection of As(III) in groundwater

Manuscript ID: RSOS-201500

The editor assigned to your manuscript has now received comments from reviewers. We would like you to revise your paper in accordance with the referee and Subject Editor suggestions which can be found below (not including confidential reports to the Editor). Please note this decision does not guarantee eventual acceptance.

Please submit your revised paper before 15-Oct-2020. Please note that the revision deadline will expire at 00.00am on this date. If we do not hear from you within this time then it will be assumed that the paper has been withdrawn. In exceptional circumstances, extensions may be possible if agreed with the Editorial Office in advance. We do not allow multiple rounds of revision so we urge you to make every effort to fully address all of the comments at this stage. If deemed necessary by the Editors, your manuscript will be sent back to one or more of the original reviewers for assessment. If the original reviewers are not available we may invite new reviewers.

RSC Associate Editor:
Comments to the Author:
(There are no comments.)

RSC Subject Editor:
Comments to the Author:
(There are no comments.)

Reviewers' Comments to Author:
Reviewer: 1

Comments to the Author(s)
Manuscript Title: Portable smartphone integrated paper sensors for fluorescence detection of As(III) in groundwater
Manuscript ID: RSOS-201500

The manuscript reports fabrication of Cu nanoclusters and their applicability for As(III) analysis by utilising integrative paper-based fluorescence detection. The authors did a lot of experiments, but there are some points should be clarified.

- 1- The authors mentioned that, "glutathione is not only a reductive agent but also a protective agent,.....", suitable references should be added.
- 2- How the authors control the Cu nanoclusters loaded on the paper? The repeatability of this sensor should be discussed.
- 3- Figure 2b should be corresponding to the variation of As(III) concentrations in Fig.2a, The X axis should be from 0.0 - 2 μ M.
- 4- The concentrations of As(III) should be legend in Fig2a.
- 5- It was found that, the detection limit using fluorescence was 2.93 nM, while increased to 92.96 nM using smartphone APP. The authors should compare the sensitivity in Figs. 2b and Fig. 4b and give explanation.
- 6- It is surprising the proposed method have a good accuracy compared to HPLC-ICP-MS as shown in Fig. 5.
- 7- The manuscript need English revision.

Reviewer: 2

Comments to the Author(s)

Level of arsenic contamination in groundwater should be strictly monitored due to their close relationship with human health and environmental safety. The manuscript describes a new material for the detection of arsenite with high sensitivity and selectivity. The detection limit of this sensor toward arsenite is quite low with a relatively wide range detection. In addition, the paper sensors were also developed and the optical signal can be quickly read by a common smart phone. Moreover, arsenite concentrations in real water samples can be determined by the as-prepared sensor. In summary, the manuscript can be accepted after minor revisions.

1 What are the advantages of Cu nanoclusters for micropollutants detection, e.g. arsenite detection?

2 In the experimental section, authors should address all of metal salts that they used in this experiment.

3 In the results and discussion, authors reported the size of the prepared Cu nanoparticles are spherical with about 3 nm. Please add standard deviation of the particle size.

4 Please clarify the operation procedure of arsenite determination method by a common smartphone equipped with Color Detector APP.

5 There are some minor errors in grammar and spelling. Authors should check and revise the manuscript more carefully.

Author's Response to Decision Letter for (RSOS-201500.R0)

See Appendix A.

RSOS-201500.R1 (Revision)

Review form: Reviewer 1

Is the manuscript scientifically sound in its present form?

Yes

Are the interpretations and conclusions justified by the results?

Yes

Is the language acceptable?

Yes

Do you have any ethical concerns with this paper?

No

Have you any concerns about statistical analyses in this paper?

No

Recommendation?

Accept as is

Comments to the Author(s)

The revised version is enhanced and suitable for publication in RSOS.

Review form: Reviewer 2

Is the manuscript scientifically sound in its present form?

Yes

Are the interpretations and conclusions justified by the results?

Yes

Is the language acceptable?

Yes

Do you have any ethical concerns with this paper?

No

Have you any concerns about statistical analyses in this paper?

Yes

Recommendation?

Accept as is

Comments to the Author(s)

I have looked through the revision made by the authors and concluded that the authors have substantially improved their manuscript. it can be accepted for publication.

Decision letter (RSOS-201500.R1)

Dear Dr Li:

Title: Portable smartphone integrated paper sensors for fluorescence detection of As(III) in groundwater

Manuscript ID: RSOS-201500.R1

It is a pleasure to accept your manuscript in its current form for publication in Royal Society Open Science. The chemistry content of Royal Society Open Science is published in collaboration with the Royal Society of Chemistry.

Yours sincerely,

Dr Laura Smith

Publishing Editor, Journals

RSC Associate Editor:
Comments to the Author:
(There are no comments.)

RSC Subject Editor:
Comments to the Author:
(There are no comments.)

Reviewer(s)' Comments to Author:
Reviewer: 2

Comments to the Author(s)
I have looked through the revision made by the authors and concluded that the authors have substantially improved their manuscript. it can be accepted for publication.

Reviewer: 1

Comments to the Author(s)
The revised version is enhanced and suitable for publication in RSOS.

Appendix A

Response to Referees

Thank you very much for your mail and for encouraging us to fully address the concerns of the referees. We are very grateful to the referees for their valuable comments, which have helped us improve the quality of the manuscript. After the careful consideration of the referee's comments, all the revisions have been made to improve the quality of our manuscript. And these revisions in the revised manuscript were marked in red.

Response to comments by Referee 1:

1 The authors mentioned that, “glutathione is not only a reductive agent but also a protective agent,.....”, suitable references should be added.

Response: Thank you for your advice. We have added three references to explain the effect of glutathione on Cu nanocluster preparation ¹⁻³.

2 How the authors control the Cu nanoclusters loaded on the paper? The repeatability of this sensor should be discussed.

Response: Thanks for your valuable suggestion. For the paper sensors based on Cu nanoclusters, different fabrication parameters were strictly controlled. And their fabrication procedures were as below. Firstly, 1g of poly (vinyl alcohol) (PVA-1788) was dissolved in 20 mL of water. Then 1 mL of Cu nanoclusters solution (5 µg/mL) was added and stirred vigorously for 15 min. After that, we dropped the common filter paper into the above mixture solution and lasted for 30s. Subsequently, the filter paper was taken out and dried at room temperature for 30 min. The paper sensors were finally fabricated after cutting into a series of strips (1 cm×5 cm) for use. Then we prepared As(III) stock solutions by mixing 20 µL of As(III) with different concentrations and 1.98 mL of BR buffer solution. The fabricated paper sensors were immersed into the As(III) stock solution. After standing for 20s, they were taken out and placed under a UV lamp to shoot their fluorescence colors (**Figure 1a**). Thus, the loading amount of Cu nanoclusters on the paper can be regulated effectively by controlling the nanoclusters concentrations, immersing time etc., then fluorescence colors from the paper sensors can be strictly controlled. We have repeated the paper sensors for several times, and obtained the similar results. However, we cannot determine the As(III) concentration accurately just *via* the fluorescence color exhibition from paper sensors. In order to solve the problem, the fluorescence color of paper sensors was analyzed by a smartphone equipped with a Color Detector APP. The smartphone was placed in a holder at a fixed distance (20 cm) for color scanning, and the color parameters (R, G and B) can be recorded by the APP. R/(R+G+B) ratio of different

fluorescence color *versus* As(III) concentrations was chosen to evaluate fluorescence response (**Figure 1b**). And we can determine As(III) concentrations from the linear equation.

Figure 1 (**Figure 4** in the revised manuscript) (a) Fluorescence images of paper sensors upon the increasing addition of As(III) (From left to right: 0 μM, 0.15 μM, 0.30 μM, 0.45 μM, 0.60 μM, 0.75 μM, 0.90 μM, 1.05 μM, 1.20 μM). All the images were taken under a 365 nm UV lamp; (b) The linear relationship between the color parameters of paper sensors and As(III) concentrations.

3 Figure 2b should be corresponding to the variation of As(III) concentrations in Fig.2a, The X axis should be from 0.0 – 2 μM.

Response: Thanks. For As(III) detection, we conducted the fluorescence titration experiments of Cu nanoclusters toward As(III) under the optimal sensing conditions. Different concentrations of As(III) (0, 0.1, 0.2, 0.3, 0.4, 0.5, 0.6, 0.7, 0.8, 0.9, 1.0, 1.1, 1.2, 1.3, 1.4, 1.5, 1.6, 1.7, 1.8, 1.9 and 2.0 μM) were added into Cu nanoclusters solution. Upon the gradual addition of As(III), the fluorescence intensity of Cu nanoclusters decreased gradually, which is based on the strong interaction between As(III) and glutathione onto the surface of nanoclusters (**Figure 2a**). The fluorescence dose response of Cu nanoclusters with As(III) was also examined (**Figure 2b**). In **Figure 2b**, the X axis represented the concentrations of As(III), which were equal to the number on X axis multiplied by 20 nM. We can find that the relative fluorescence intensities $(F_0-F)/F_0$ increase dramatically with the increasing As(III) concentrations initially, and it tends to be saturated at a higher concentration. A linear fit for the linear range from 0 to 1.3 μM was presented (*inset* of **Figure 2b**). And the correlation coefficient was as high as 0.991. The detection limit of Cu nanoclusters toward As(III) is calculated to be 2.93 nM (0.22 ppb) based on the 3SD/ρ rule.

Figure 2 (**Figure 2** in the revised manuscript) (a) Fluorescence spectra of Cu nanoclusters with increasing addition of As(III) (0 to 2.0 μM) under the excitation of 380 nm, all the fluorescence measurements were conducted in BR buffer solution, (b) Plot of the fluorescence change $(F_0-F)/F_0$ versus As(III) concentrations (0 to 2 μM), and the *inset* is the liner plot of fluorescence changes versus As(III) concentrations (0 to 1.3 μM).

4 The concentrations of As(III) should be legend in Fig 2a.

Response: Thanks. We have added As(III) concentrations in **Figure 2a**. The revised figure can be seen in **Figure 2** (**Figure 2** in the revised manuscript).

5 It was found that, the detection limit using fluorescence was 2.93 nM, while increased to 92.96 nM using smartphone APP. The authors should compare the sensitivity in Figs. 2b and Fig. 4b and give explanation.

Response: Thanks for your valuable suggestion. The detection limit using fluorescence was calculated to be 2.93 nM (0.22 ppb) (**Figure 2b**). Comparing with some reported fluorescence sensors and As(III) permitted level in drinking water, the detection limit was much lower⁴⁻¹¹. Then we loaded Cu nanoclusters on common filter paper to fabricate the paper sensors. Then different concentrations of As(III) stock solutions were immersed and obtained different fluorescence colors under UV lamp. Their parameters, such as red (R), green (G), blue (B) can be present^{12, 13}. $R/(R+G+B)$ ratio of different color versus As(III) concentrations was chosen to evaluate fluorescence response. A calibration curve with the coefficient of 0.992 between the fluorescence color changes and As(III) concentrations ranging from 0 to 1.2 μM was constructed founded (**Figure 4b**). The lowest detectable concentration of the paper sensor is approximately 92.96 nM (6.98 ppb), which is also lower than the As(III) permitted level in drinking water. Obviously, the detection limit from the paper sensors is higher than that from the fluorescence detection method. Different factors, such as the tight loading of Cu nanoclusters on the paper, color parameter collection sensitivity, UV lamp power, exposure time etc. would decrease the detection sensitivity of paper sensors toward

As(III)^{14, 15}. Nevertheless, the method of integrating smartphone with fluorescence paper sensor can complete the visual, on-site and quantitative detection of As(III).

6 It is surprising the proposed method have a good accuracy compared to HPLC-ICP-MS as shown in Fig. 5.

Response: Thanks. We have evaluated the practical As(III) detection ability of smartphone integrated paper sensor. The natural groundwater samples from some high-arsenic groundwater areas in China, such as Datong basin, Xiantao Jiangnan Plain, Foshan Pearl River Delta, and Kuitun Xinjiang were collected and treated by centrifugation, filtration and pH adjustment. As(III) concentrations in these natural groundwater samples were determined by HPLC-ICP-MS and our developed smartphone integrated paper sensors in triplicate and the average values were presented with standard deviation. The higher Pearson correlation coefficient of 0.998 and lower p-value of 0.001 between the two As(III) concentrations determination methods indicated that the smartphone integrated paper sensors possessed the similar performance to HPLC-ICP-MS on arsenic speciation determination. The experimental results indicated that As(III) concentrations from smartphone integrated paper sensors is reliable and can be applied for As(III) detection in the real groundwater samples.

7 The manuscript need English revision.

Response: Thanks. We have checked and revised the manuscript according to your advice.

Response to comments by Referee 2:

1 What are the advantages of Cu nanoclusters for micropollutants detection, e.g. arsenite detection?

Response: Thanks. As known, the fluorescence method is a suitable candidate for micropollutants detection due to their higher sensitivity, selectivity, shorter analysis time, and clearer visual ability¹⁶. In this type of sensors, the signaling unit may consist of fluorescent organic molecules or luminescent nanomaterials¹⁷. The use of quantum dots as sensing materials has been questioned because many of these nanoparticles are composed of heavy metals¹⁸. Metal nanoclusters were extensively studied due to their high luminescence originated from their quantum confinement. Typically, gold and silver are used to develop luminescent nanoclusters to for ions, organic molecules etc. detection¹⁹⁻²³. However, gold and silver precursors are very expensive, so there is a great interest in developing economic and biocompatible new metal nanoclusters. The non-noble metal Cu nanoclusters can avoid the shortcoming of quantum dots and gold, silver nanoclusters, then the study of Cu nanoclusters have also attracted extensive attentions due to their earth-

abundance, lower cost, outstanding fluorescence property, easy-to-use decoration etc.²⁴⁻²⁶ In this manuscript, we prepared the water soluble Cu nanoclusters for sensitive and selective detection of As(III).

2 In the experimental section, authors should address all of metal salts that they used in this experiment.

Response: Thanks. We have address all of the metal salts in this experiment. And the revision is in the revised Supporting information. Then sentence “The interfering anions NO_3^- , CO_3^{2-} , HCO_3^- , SO_4^{2-} , $\text{C}_2\text{O}_4^{2-}$, PO_4^{3-} , BrO_3^- , ClO_4^- , F^- , Cl^- , Br^- , I^- with sodium ion as counter cations, and interfering cation Na^+ , K^+ , Al^{3+} , Fe^{3+} , Fe^{2+} , Cu^{2+} , Zn^{2+} , Cd^{2+} , Hg^{2+} , Ca^{2+} , Mg^{2+} and Pb^{2+} with nitrate as counter anions were obtained from Sinopharm Chemical Reagent Co. Ltd.”.

3 In the results and discussion, authors reported the size of the prepared Cu nanoparticles are spherical with about 3 nm. Please add standard deviation of the particle size.

Response: Thanks. TEM image was obtained to study the morphology and particle size distributions of Cu nanoclusters. As shown in **Figure 1b** in the revised manuscript, the prepared Cu nanoparticles are spherical with about 3.0 ± 0.2 nm and finely disseminated with no clear aggregations. The data of size distribution analysis of Cu nanoclusters from Malvern potential instrument is consistent with the TEM analysis (Figure S8 in the revised manuscript).

4 Please clarify the operation procedure of arsenite determination method by a common smartphone equipped with Color Detector APP.

Response: Thanks. The paper sensors were firstly fabricated and then immersed in As(III) stock solutions of different concentrations (0 μM , 0.15 μM , 0.30 μM , 0.45 μM , 0.60 μM , 0.75 μM , 0.90 μM , 1.05 μM , 1.20 μM). After standing for 20s, the paper sensors were taken out and placed under a UV lamp to shoot their fluorescence colors (**Figure 4a** in the revised manuscript). The fluorescence color of paper sensors under UV lamp was analyzed by a smartphone equipped with a Color Detector APP. The smartphone was placed in a holder at a fixed distance (20 cm) for color scanning, and the color parameters (R, G and B) can be recorded by the APP. R/(R+G+B) ratio of different color *versus* As(III) concentrations was chosen to evaluate fluorescence response. A calibration curve with the coefficient of 0.992 between the fluorescence color changes and As(III) concentrations ranging from 0 to 1.2 μM was constructed founded (**Figure 4b** in the revised manuscript). As(III) concentrations in different water samples were then determined by comparing with the calibration curve.

5 There are some minor errors in grammar and spelling. Authors should check and revise the manuscript more carefully.

Response: Thanks. We have checked and revised the manuscript according to your advice.

Reference

- 1 Y. Luo, H. Miao and X. Yang, Glutathione-stabilized Cu nanoclusters as fluorescent probes for sensing pH and vitamin B1. *Talanta*, 2015, **144**, 488-495.
- 2 C. Wang, D. Zhang, L. Xu, Y. Jiang, F. Dong, B. Yang, K. Yu and Q. Lin, A Simple Reducing Approach Using Amine To Give Dual Functional EuSe Nanocrystals and Morphological Tuning. *Angew. Chem. Int. Ed.*, 2011, **50**, 7587-7591.
- 3 Z. Cai, R. Zhu, S. Pang, F. Tian and C. Zhang, One- step Green Synthetic Approach for the Preparation of Orange Light Emitting Copper Nanoclusters for Sensitive Detection of Mercury(II) Ions. *ChemistrySelect*, 2020, **5**, 3682-3687.
- 4 S. Roy, G. Palui and A. Banerjee, The as-prepared gold cluster-based fluorescent sensor for the selective detection of As(III) ions in aqueous solution. *Nanoscale*, 2012, **4**, 2734-2740.
- 5 Y.Y. Zhang, Y. Shi, X.L. Zhao, Y.P. Luo, C.Y. Deng and S.H. Si, Simultaneous Determination of Trace Arsenic and Lead by Stripping Voltammetry in Groundwater of Mines. *Anal. Sci.*, 2012, **4**, 519-522.
- 6 N. Butwong, T. Noipa, R. Burakham, S. Srijaranai and W. Ngeontae, Determination of arsenic based on quenching of CdS quantum dots fluorescence using the gas-diffusion flow injection method. *Talanta*, 2011, **85**, 1063-1069.
- 7 T. Alizadeh and M. Rashedi, Synthesis of nano-sized arsenic-imprinted polymer and its use as As(III) selective ionophore in a potentiometric membrane electrode: Part 1. *Anal. Chim. Acta.*, 2014, **843**, 7-17.
- 8 M. Talat, O. Prakash and S.H. Hasan, Enzymatic detection of As(III) in aqueous solution using alginate immobilized pumpkin urease: Optimization of process variables by response surface methodology. *Bioresour. Technol.*, 2009, **100**, 4462-4467.
- 9 M. Tang, G. Wen, A. Liang and Z. Jiang, A simple and sensitive resonance Rayleigh scattering method for determination of As(III) using aptamer-modified nanogold as a probe. *Luminescence*, 2014, **29**, 603-608.
- 10 M.M. Hossain, M.M. Islam, S. Ferdousi, T. Okajima and T. Ohsaka, Anodic Stripping Voltammetric Detection of Arsenic(III) at Gold Nanoparticle-Modified Glassy Carbon Electrodes Prepared by Electrodeposition in the Presence of Various Additives. *Electroanalysis*, 2008, **20**, 2435-2441.
- 11 Y. Wu, F. Wang, S. Zhan, L. Liu, Y. Luo and P. Zhou, Regulation of hemin peroxidase catalytic

- activity by arsenic-binding aptamers for the colorimetric detection of arsenic(III). *RSC. Adv.*, 2013, **3**, 25614-25619.
- 12 J.X.H. Wong, F.S.F. Liu and H.Z. Yu, Mobile APP-based quantitative scanometric analysis. *Anal. Chem.*, 2014, **86**, 11966-11971.
- 13 L. Li, Z.G. Liu, H. Zhang, W.Q. Yue, C.W. Li and C.Q. Yi, A point-of-need enzyme linked aptamer assay for Mycobacterium tuberculosis detection using a smartphone. *Sens. Actuators B*, 2018, **254**, 337-346.
- 14 R. Sun, X. Huo, H. Lu, S. Feng, D. Wang and H. Liu, Recyclable fluorescent paper sensor for visual detection of nitroaromatic explosives. *Sens. Actuators B*, 2018, **265**, 476-487.
- 15 H. Wang, L. Da, L. Yang, S. Chu, F. Yang, S. Yu and C. Jiang, Colorimetric fluorescent paper strip with smartphone platform for quantitative detection of cadmium ions in real samples. *J. Hazard. Mater.*, 2020, **392**, 122506-122513.
- 16 T.I. Kim and Y. Kim, A water indicator strip: instantaneous fluorogenic detection of water in organic solvents, drugs, and foodstuffs. *Anal. Chem.*, 2017, **89**, 3768-3772.
- 17 Z. Chen, S. Qian, J. Chen and X. Chen, Highly fluorescent gold nanoclusters based sensor for detection of quercetin. *J. Nanopart. Res.*, 2012, **14**, 1264-1271.
- 18 S.J. Soenen, B.B. Manshian, T. Aubert, U. Himmelreich, J. Demeester, S.C. De Smedt, Z. Hens and K. Braeckmans, Cytotoxicity of cadmium-free quantum dots and their use in cell bioimaging. *Chem. Res. Toxicol.*, 2014, **27**, 1050-1059.
- 19 Y. Lin and W. Tseng, Ultrasensitive sensing of Hg^{2+} and CH_3Hg^+ based on the fluorescence quenching of lysozyme type VI-stabilized gold nanoclusters. *Anal. Chem.*, 2010, **82**, 9194-9200.
- 20 J.A. Annie Ho, H.C. Chang and W.T. Su, DOPA-mediated reduction allows the facile synthesis of fluorescent gold nanoclusters use as sensing probes for ferric ions. *Anal. Chem.*, 2012, **84**, 3246-3253.
- 21 G. Zhang, Y. Qiao, T. Xu, C. Zhang, Y. Zhang, L. Shi, S. Shuang and C. Dong, Highly selective and sensitive nanoprobe for cyanide based on gold nanoclusters with red fluorescence emission. *Nanoscale*, 2015, **7**, 12666-12672.
- 22 X. Deng, X. Huang and D. Wu, Förster resonance-energy-transfer detection of 2,4,6-trinitrophenol using copper nanoclusters. *Anal. Bioanal. Chem.*, 2015, **407**, 4607-4613.
- 23 N. Enkin, E. Sharon, E. Golub and I. Willner, Ag nanoclusters/DNA hybrids: functional modules for the detection of nitroaromatic and RDX explosives. *Nano Lett.*, 2014, **14**, 4918-4922
- 24 Z. Qing, T. Qing, Z. Mao, X. He, K. Wang, Z. Zou, H. Shi and D. He, dsDNA-specific fluorescent copper nanoparticles as a "green" nano-dye for polymerization-mediated biochemical analysis. *Chem. Commun.*, 2014, **50**, 12746-12748.
- 25 Y. Lu and W. Chen, Sub-nanometre sized metal clusters: from synthetic challenges to the unique property discoveries. *Chem. Soc. Rev.*, 2012, **41**, 3594-3623.

26 X. Tian, L. Liu, Y. Li, C. Yang, Z. Zhou, Y. Nie and Y. Wang, Nonenzymatic electrochemical sensor based on CuO-TiO₂ for sensitive and selective detection of methyl parathion pesticide in ground water. *Sens. Actuators B*, 2018, **256**, 135-142.